# Effects of Spatial Reference Frames, Map Dimensionality, and Navigation Modes on Spatial Orientation Efficiency

Hongyun Guo [1], Nai Yang [2], Zhong Wang [3] and Hao Fang [1,*]

1   School of Art & Design, Wuhan Institute of Technology, Wuhan 430205, China; guohongyun@wit.edu.cn
2   School of Geography and Information Engineering, China University of Geosciences (Wuhan),
    Wuhan 430074, China; yangnai@cug.edu.cn
3   School of Arts and Communication, China University of Geosciences (Wuhan), Wuhan 430074, China;
    wangzhong0820@cug.edu.cn
*   Correspondence: fanghao@wit.edu.cn

**Abstract:** How can the interactive mode of a map be optimized to facilitate efficient positioning and improve cognitive efficiency? This paper addresses this crucial aspect of map design. It explores the impact of spatial reference frames, map dimensionality, and navigation modes on spatial orientation efficiency, as well as their interactions, through empirical eye-movement experiments. The results demonstrate the following: (1) When using a 2D fixed map in an allocentric reference frame, participants exhibit a high correct rate, a low cognitive load, and a short reaction time. In contrast, when operating within an egocentric reference frame using a 2D rotating map, participants demonstrate a higher correct rate, a reduced cognitive load, and a quicker reaction time. (2) The simplicity of 2D maps, despite their reduced authenticity compared to 3D maps, diminishes users' cognitive load and enhances positioning efficiency. (3) The fixed map aligns more closely with the cognitive habits of participants in the allocentric reference frame, while the rotating map corresponds better to the cognitive habits of participants in the egocentric reference frame, thereby improving their cognitive efficiency. This study offers insights that can inform the optimization design of spatial orientation efficiency.

**Keywords:** spatial reference frame; navigation mode; map dimensionality; spatial orientation efficiency

## 1. Introduction

Navigational maps, meticulously crafted to aid individuals in identifying locations and planning routes, are essential in contemporary society [1]. With advancements in geographic information science, these maps can precisely depict Earth's geographic features, such as road networks, landmarks, and landforms, thus offering users crucial navigational references. The efficacy of users in spatial orientation tasks is considerably influenced by the interaction methods employed in navigational maps. The fundamental goal of navigational-map design is to ascertain the most effective method of presenting the map to users, ensuring accurate location identification and enhanced cognitive efficiency [2].

Map dimensionality refers to the spatial representation on a navigational map. Specifically, it indicates whether the map portrays information in two dimensions (2D) or three dimensions (3D) [3]. Depending on this aspect, navigational maps are categorized into 2D and 3D varieties. Regarding navigation approaches, maps are further divided into static and rotating types. Static maps remain north-oriented, with the user icon rotating slightly as the user's direction changes. In contrast, rotating maps dynamically adapt to the user's forward orientation, aligning the map's top with the user's current heading [4].

Although prior studies have explored the spatial orientation efficiency of different navigational maps, their conclusions have been varied and sometimes contradictory [5–8]. Limited research has simultaneously examined the impact of both map dimensionality and navigation mode on spatial orientation efficiency. Additionally, individuals often depend

on specific reference systems for discerning the direction and position of targets in space. It has been observed that different individuals favor distinct spatial reference systems in their everyday navigation [9]. Montello observed that, during navigation, an individual's reference frame can alternate between egocentric and allocentric frames [10]. The egocentric frame is based on personal viewpoints and body orientation, while the allocentric frame focuses on external cues and the relationships between objects in the environment [11]. Research has underscored that completing spatial orientation tasks with maps involves mental rotation and target search [12]. Beyond map dimensionality and navigation mode, these reference systems might also affect the mental rotation process.

Building upon existing research, this article introduces spatial reference systems as an individual characteristic factor in digital map navigation. It delves into the interplay among spatial reference frames, map dimensionality, and navigation mode concerning spatial orientation efficiency. Through empirical research, this study offers design principles to inform future navigational map design practices.

## 2. Literature Review

### 2.1. Research on the Influence of Map Dimensionality on Spatial Orientation Efficiency

Numerous studies have indicated that map dimensionality has an impact on spatial orientation efficiency. Oulasvirta et al.'s field experiment revealed that 2D maps minimized users' cognitive load and informed them about inflection points and decision points in advance, which are crucial junctures in navigation [13]. Partala et al. [14] found that while 3D maps engaged users and aided quick landmark identification, the associated cognitive load made many favor 2D visual maps. Lei et al. [15] utilized eye-movement indicators in a visual search task to contrast the effects of 3D and 2D maps on users' visual attention. Their findings indicated broader gaze ranges in 2D maps, facilitating quicker browsing. Liao's eye-tracking experiments on path-finding tasks showed that, while 3D maps led to longer response times, they were more effective in certain tasks. With 2D maps, users acquired more spatial knowledge, but 3D maps were preferred for decision making at intricate junctions [5]. Dong et al.'s eye movement experiments underscored the high cognitive efficiency and low cognitive load of 2D maps, while 3D maps excelled in self-positioning and path decision making [6].

Based on previous studies, we have found that in map navigation tasks, participants using 3D maps can be more helpful in quickly searching for required information in complex scenarios. However, 2D maps impose a lower cognitive load, allowing participants to demonstrate greater spatial cognitive efficiency in simpler tasks.

### 2.2. Research on the Influence of Navigation Mode on Spatial Orientation Efficiency

Various studies have highlighted the unique cognitive impacts of navigation methods, particularly between fixed and rotating maps. Aretz's comparative study on pilots navigating using two types of maps elucidated the benefits of rotating maps in their alignment with the participant's reference frame. Conversely, fixed maps have the advantage of consistent reference frames, regardless of rotation. Importantly, both map types necessitate visual guidance for auxiliary navigation design [4]. In subsequent research, Aretz identified a superior navigation efficiency with rotating maps for steering tasks, while fixed maps, due to their increased cognitive load, took longer [16]. Wickens, on the other hand, asserted that the consistent reference frame of fixed maps results in users experiencing a reduced cognitive load, while rotating maps excel in tasks requiring spatial orientation [17]. Despite the disagreement on cognitive loads associated with different map navigation methods, there is consensus about the higher efficiency of rotating maps in spatial positioning tasks. Mou Shu's research highlighted that under simple backgrounds, rotating maps outperformed fixed maps. However, as background complexity increased, the performance difference between the two types diminished [18]. Another important observation by Rodes et al. was the superior performance of fixed maps in map-reconstruction tasks. In contrast, rotating maps were more adept in guidance and route-following tasks. It is also essential to consider

that the efficacy of map navigation does not solely rely on the map format but is influenced by the individual's spatial cognitive abilities [19]. A maze experiment conducted by Robbi Sluter and his team revealed no significant differences in user reaction times between fixed and rotating maps [20].

Diving deeper into the internal mechanism of spatial cognition for both map types, Levine introduced the "adjustment effect" mechanism. This proposes that when the map's reference frame is aligned with the individual's self-reference frame, there is no need for psychological rotation, leading to quicker self-positioning tasks and improved navigation efficiency [21]. However, Tamura's discovery of the "geographical indication effect" suggests the boundaries of the adjustment effect. Specifically, as map complexity increases, due to added geographical and background information, the adjustment effect starts to wane or even disappear. The reasoning behind this is that post map rotation, the changing background and landmarks necessitate users to search and locate targets afresh, making the process time-consuming [22].

To summarize, while many believe that rotating maps have an edge in positioning efficiency over fixed maps, others argue that personal preferences, influenced by individual spatial reference frames, play a pivotal role. Some might lean toward fixed maps, due to their ability to provide a more holistic understanding of spatial layouts. The inherent time-saving nature of rotating maps can be attributed to the adjustment effect. However, as map symbols and background intricacies augment, this effect begins to reduce, giving way to the geographical-indication effect. This research hypothesizes that, concerning navigation mode, rotating maps seem to exhibit better spatial positioning efficiency than fixed maps, though this efficiency is intertwined with factors such as map background, task intricacy, and the individual's spatial cognitive prowess.

### 2.3. Research on Spatial Reference Frame in the Field of Map Navigation

In the field of map navigation, understanding the selection and application of spatial reference frames remains a pivotal research topic. Montello [23] elucidated that user preferences for spatial reference frames during navigation can differ significantly. Some might prefer maps that dynamically adjust based on their current direction, while others opt for traditional static maps with a north-up orientation. Further building on this foundation, Shelton and McNamara [9] delved into the applicability of egocentric versus allocentric frames in spatial tasks. They found that the efficacy of these frames is contingent on the nature of the task and individual user differences. Specifically, one frame might be more advantageous than the other in certain spatial tasks, such as navigation or object localization.

Exploring the cognitive intricacies, some scholars have investigated the connection between spatial reference frames and mental rotation. When navigating unfamiliar terrains with a map not aligned with one's viewpoint, an individual might resort to mentally rotating the map's content, blending an allocentric representation into their current egocentric perspective [24]. In tasks of object recognition from unfamiliar viewpoints, the cognitive mechanism of mental rotation becomes intertwined with an individual's internal spatial reference frame. This often requires them to mentally adjust the object to correspond with a known orientation [25]. Golledge et al. [26] further emphasized the significance of mental rotation in spatial navigation, especially within intricate and unfamiliar environments. Moreover, Hegarty [27] shed light on the relationship between spatial abilities and navigation tasks. He proposed that individual variations in these capabilities could be a determinant in performance metrics across diverse spatial tasks. The role of perspective-taking is also worthy of attention, as it encompasses the cognitive capacity to perceive or conceptualize a situation from an alternative viewpoint [28]. Though bearing resemblance to mental rotation abilities, perspective-taking influences spatial cognition tasks in its unique ways, possibly attributed to the switching of reference frames.

Li Yu's investigations revealed interesting outcomes about the effect of spatial reference frames on mental rotation. It was discerned that the performance of participants

utilizing an egocentric reference frame during orientation determination remains robust even during mental rotation. However, those relying on an allocentric reference frame showed a marked decline in performance during such tasks [29]. On the other hand, Liang Le's studies painted a slightly different picture. They documented that the individuals within the allocentric reference frame group outperformed their counterparts, suggesting heightened spatial-orientation capabilities [30]. These research outcomes denote that the choice of different reference frames might influence users' mental rotations. Still, potentially due to other contributing factors, participants using an allocentric reference frame may demonstrate enhanced spatial cognitive performance.

In conclusion, the spatial reference frame, as a significant factor in individual spatial cognition, is closely related to psychological rotation in the process of spatial orientation. Therefore, this paper includes it in the scope of research and combines it with navigation mode and map dimensionality to discuss their combined influence on spatial orientation efficiency. This provides a basis for designing and applying maps accordingly.

### 2.4. Hypothesis

According to the previous literature, this study puts forward the following assumptions:

**Hypothesis 1 (H1).** *In map navigation tasks, individuals exhibit higher spatial orientation efficiency in 2D maps than in 3D maps.*

**Hypothesis 2 (H2).** *In map navigation tasks, individuals demonstrate greater spatial orientation efficiency with rotating maps than with fixed maps.*

**Hypothesis 3 (H3).** *In map navigation tasks, individuals with an allocentric reference frame demonstrate higher spatial orientation efficiency than those with an egocentric reference frame.*

## 3. Materials and Methods

### 3.1. Pre-Experiment: Participant Screening Experiment

The arrangement paradigm designed by Levinson has been empirically validated to discern the reference frames adopted by participants [31,32]. Leveraging this paradigm, we aimed to select participants who predominantly utilized either egocentric or allocentric reference frames for the study. The screening experiment was conducted in a laboratory with two tables positioned to the east and west, respectively. Three symmetrical dragonfly toys were placed in a linear arrangement from north to south on one of the tables. (Refer to Figure 1, where the smiling face next to the table showcases the orientation and position of the participant).

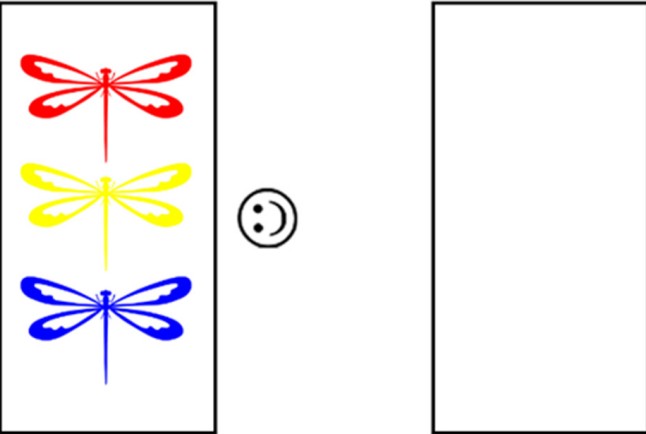

**Figure 1.** Demonstration diagram of animal testing.

Each participant underwent the test individually. The experimenter presented the dragonflies on one of the tables, aligned from north to south. Participants were instructed: "Please remember the spatial relationships between the dragonfly toys. Inform me once you've committed the arrangement to memory". No time limit was imposed. Following their confirmation, they were guided to rotate 180 degrees to face the opposing table. The toys were then handed to the participant to reproduce the initial arrangement. The directive was: "Replicate the arrangement of the dragonflies based on your memory". Directional words such as "east, south, west, north, up, down, left, right", were deliberately avoided, and neutral terms like "here" and "there" were employed when required. Participants believed they were partaking in a memory task, with no hint toward making spatial judgments. This design was intended to capture spontaneous spatial decisions, devoid of overt influence. Potential questions from participants related to the task were not entertained post initiation, ensuring that their spatial decisions remained uninfluenced.

The underlying logic of the experiment was as follows: the dragonflies on the original table had their heads facing upwards, which also pointed to the right of the participant. If a participant consistently recreated the scene by placing the dragonflies with their heads facing upwards, it indicated the use of an allocentric reference frame. If a participant placed the dragonflies with their heads facing downwards (toward the right) when recreating the scene, it signaled the use of an egocentric reference frame (as illustrated in Figure 2).

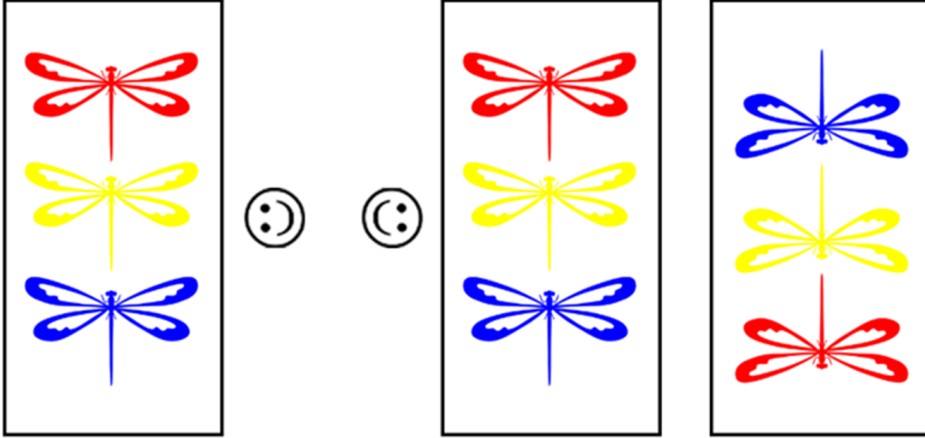

**Figure 2.** Animal test results of two different participants.

### 3.2. Participants

Participants were students from China University of Geosciences, aged 18 to 25. A pre-experiment yielded 61 participants: 31 in the allocentric and 30 in the egocentric reference frame groups.

### 3.3. Experimental Design

The experiment utilized a three-factor mixed design involving the following factors: map dimensionality (2 levels: 3D map and 2D map), spatial reference frame (2 levels: allocentric reference frame and egocentric reference frame), and navigation mode (2 levels: rotating map and fixed map). The spatial reference frame was considered an independent group variable, while the navigation mode and map dimensionality were repeated measures variables of the participants. The dependent variables included the response time and the accuracy rate of the participants when completing the spatial orientation task, as well as eye-movement indices such as fixation duration, pupil diameter, and average saccade amplitude.

Specifically, the accuracy rate represented the probability of participants correctly completing the navigational positioning task, determined by collecting mouse clicks within a predefined area corresponding to an arrow. The reaction time referred to the time taken by participants to complete the navigation and positioning task. Fixation duration referred

to the total time that a participant gazed at all areas of interest on a map. It provided an objective measure of the cognitive resources expended by the participant. The pupil diameter, as a measure of cognitive load, could indicate the level of cognitive effort exerted by the participants. Saccade amplitude referred to the angle of eye movement from one fixation point to another. The average saccade amplitude was calculated as the total saccade amplitude divided by the number of saccades [8,33–35].

### 3.4. Materials

The experimental materials consisted of 32 video clips—specifically, 8 each of 2D fixed maps, 2D rotating maps, 3D fixed maps, and 3D rotating maps. Within these videos, there was a red target point and a blue moving point. Depending on the initial position of the red target point, which could be in any of the four corners of the map, combined with the blue moving point turning either left or right, there were 8 possible scenarios. For instance, one scenario could be the red target point in the top-left corner with the blue moving point turning right.

The experimental video materials were displayed on a computer screen. When the map video was shown, a blue moving point entered from the bottom and maintained a certain speed toward the center of the map. Upon reaching the central turning point of the map, its direction changed. For the rotating map, the user perceived the blue moving point's direction as unchanged, but the map's background rotated by 90 degrees. In the case of the fixed map, the map background remained stationary, and the user perceived the moving point itself to have rotated by 90 degrees. After the moving point changed direction, the participant's task was to determine the position of the red target point relative to the blue moving point and click the arrow that represented this direction, using a mouse. The initial and final frames of the video materials are shown in Figure 3. The specific video experimental materials can be found in our Supplementary Materials.

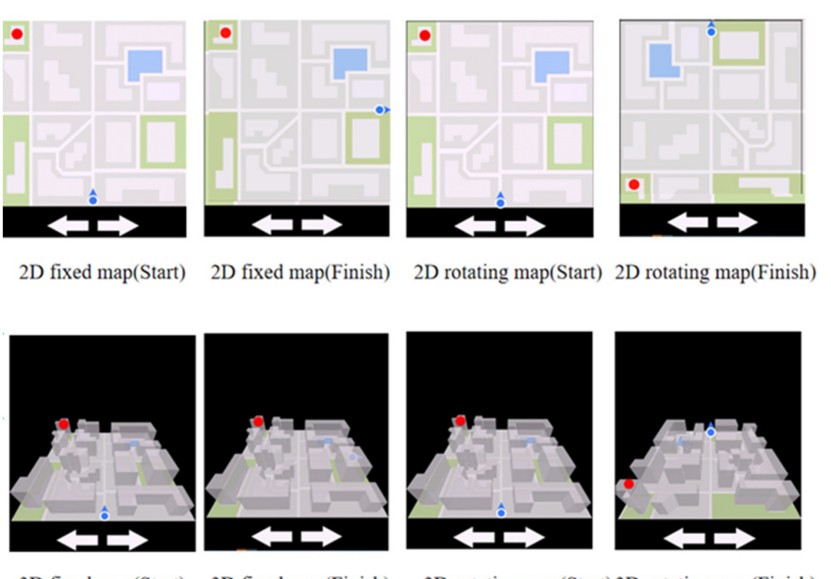

**Figure 3.** Four experimental materials.

### 3.5. Experimental Equipment and Procedures

#### 3.5.1. Equipment

In this study, the eye tracker equipment used was the Tobii IX 2-60, with a sampling rate of 60 Hz. The experimental materials were presented on the screen of a Lenovo U410 notebook computer, which had a screen refresh rate of 60 Hz and a screen resolution of 1366 × 768. The experimental software utilized was Tobii Studio 3.4.8.

3.5.2. Procedures

The experiment consisted of two stages: the practice stage and the formal experiment. Before the study began, the examiner provided the participants with relevant instructions and precautions. The eye-movement calibration stage followed, and once the calibration was completed, the experimental process began. The first page displayed on the screen contained the experimental instructions, explaining the specific tasks and operation methods to the participants. The experimental instructions were as follows: "Thank you for participating in this experiment. In each upcoming map, there will be a blue moving point representing your direction of movement, as well as a red target point. Upon the turning event of the moving point, please swiftly determine the orientation of the target point relative to the moving point, and click on the arrow representing that direction with your mouse". After understanding the instructions, the participants clicked the mouse to proceed to the practice stage. In the practice stage, four maps were presented to the participants, corresponding to different navigation scenarios, allowing the participants to familiarize themselves with the correct operations for each scenario. Once the practice stage was completed and there were no further doubts or questions, the participants proceeded to the formal experiment stage. If any doubts arose, the main examiner was available to provide clarification. Participants could only respond based on the video materials they watched and could not adjust the viewpoint. The formal experiment consisted of 32 test tasks, with each experimental material randomly presented in a sequential manner.

## 4. Data Analysis

### *4.1. Behavior Indicators*

#### 4.1.1. Accuracy Rate

Descriptive statistics for the accuracy rate are presented in Table 1.

**Table 1.** Descriptive statistics of accuracy rate (%).

| Map Dimensionality | Navigation Mode | Spatial Reference Frame | Average Value | Standard Deviation |
|---|---|---|---|---|
| 2D map | Fixed map | Allocentric reference frame | 96.774 | 7.194 |
| | | Egocentric reference frame | 90.833 | 14.656 |
| | | overall mean | 93.852 | 11.775 |
| 2D map | Rotating map | Allocentric reference frame | 97.984 | 5.680 |
| | | Egocentric reference frame | 99.583 | 2.282 |
| | | overall mean | 98.770 | 4.393 |
| 3D map | Fixed map | Allocentric reference frame | 92.339 | 13.570 |
| | | Egocentric reference frame | 91.667 | 14.056 |
| | | overall mean | 92.008 | 13.699 |
| 3D map | Rotating map | Allocentric reference frame | 95.968 | 7.491 |
| | | Egocentric reference frame | 100.000 | 0.000 |
| | | overall mean | 97.951 | 5.674 |

The data were found to meet the criteria of normal distribution and homogeneity of variance ($p > 0.05$). A repeated measures ANOVA conducted revealed the following:

The main effect of navigation mode was significant, $F(1,60) = 13.154$, $p < 0.001$. Accuracy for the fixed map was significantly lower than for the rotating map. The main effect of map dimensionality was not significant, $F(1,60) = 3.354$, $p = 0.072$. The main effect of spatial reference frame was not significant, $F(1,60) = 0.008$, $p = 0.928$.

There was a significant interaction between map dimensionality and spatial reference frame, $F(1,60) = 7.353$, $p < 0.01$. Further simple effects analysis (see Figure 4) revealed a significant difference in accuracy between 2D and 3D maps under the allocentric reference

frame, $p < 0.01$, with the 3D map having a lower accuracy rate than the 2D map. However, under the egocentric reference frame, there was no significant difference in accuracy rate between the 2D and 3D maps, $p = 0.539$. The interaction effect between navigation method and spatial reference frame was significant, $F(1,60) = 4.104$, $p < 0.05$. Further simple effects analysis (see Figure 5) showed no significant difference between fixed and rotating map for the allocentric reference frames, $p = 0.258$. In contrast, for the egocentric reference frame, a significant difference was observed, $p < 0.001$, with the accuracy rate under the fixed map being significantly lower than under the rotating map. The interaction between map dimension and navigation method was not significant, $F(1,60) = 0.374$, $p = 0.543$. The three-way interaction was also non-significant, $F(1,60) = 0.749$, $p = 0.390$.

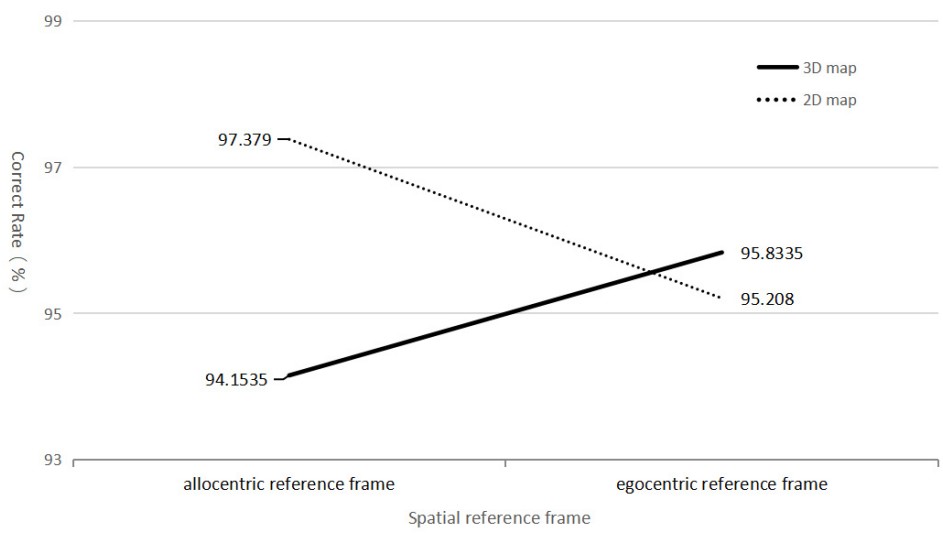

**Figure 4.** Interaction plots of spatial reference frame and map dimensionality on accuracy rate.

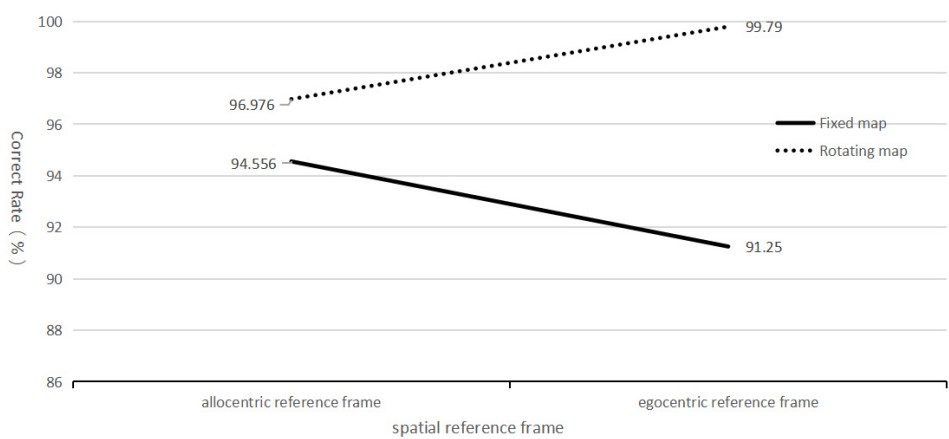

**Figure 5.** Interaction plots of spatial reference frame and navigation method on accuracy rate.

### 4.1.2. Reaction time

Descriptive statistics for the reaction time are presented in Table 2.

The data were found to meet the criteria of normal distribution and homogeneity of variance ($p > 0.05$). A repeated measures ANOVA conducted revealed the following:

The main effect of map dimensionality was significant, $F(1,60) = 17.560$, $p < 0.001$, with the reaction time for 2D maps being significantly shorter than that for 3D maps. The main effect of the spatial reference frame was significant, $F(1,60) = 21.876$, $p < 0.001$, with participants in the allocentric reference frame showing a significantly shorter reaction time than those in the egocentric reference frame. The main effect of navigation mode was not significant, $F(1,60) = 0.038$, $p = 0.847$.

**Table 2.** Descriptive statistics of reaction time (S).

| Map Dimensionality | Navigation Mode | Spatial Reference Frame | Average Value | Standard Deviation |
|---|---|---|---|---|
| 2D map | Fixed map | Allocentric reference frame | 5.203 | 0.437 |
| | | Egocentric reference frame | 5.688 | 0.341 |
| | | overall mean | 5.441 | 0.460 |
| 2D map | Rotating map | Allocentric reference frame | 5.360 | 0.291 |
| | | Egocentric reference frame | 5.594 | 0.317 |
| | | overall mean | 5.475 | 0.324 |
| 3D map | Fixed map | Allocentric reference frame | 5.273 | 0.386 |
| | | Egocentric reference frame | 5.814 | 0.333 |
| | | overall mean | 5.539 | 0.450 |
| 3D map | Rotating map | Allocentric reference frame | 5.369 | 0.307 |
| | | Egocentric reference frame | 5.633 | 0.341 |
| | | overall mean | 5.499 | 0.348 |

The interaction effect between map dimensionality and navigation mode was significant, $F(1,60) = 7.555$, $p = 0.008$, $p < 0.05$. Further simple effects analysis, illustrated in Figure 6, revealed that for fixed maps, there was a significant difference in reaction time between 2D and 3D maps, $p < 0.001$, with the reaction time for 2D maps being significantly shorter than that for 3D maps. However, for rotating maps, there was no significant difference in reaction time between 2D and 3D maps, $p = 0.223$.

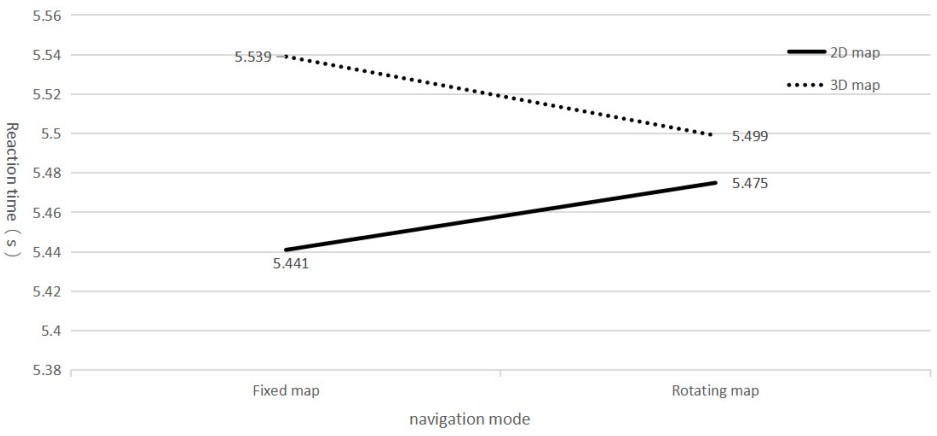

**Figure 6.** Interaction plots of navigation mode and map dimensionality on reaction time.

The interaction effect between navigation mode and spatial reference frame was significant, $F(1,60) = 19.769$, $p < 0.001$. Further simple effects analysis, as shown in Figure 7, indicated that for allocentric reference frame, there was a significant difference in reaction time between fixed and rotating maps, $p = 0.004$, $p < 0.01$, with participants in the rotating map displaying a significantly longer reaction time than those in the fixed map. For egocentric reference frame, there was a significant difference in reaction time between fixed and rotating maps, $p = 0.002$, $p < 0.01$, with participants in the fixed map having a significantly longer reaction time than those in the rotating map.

The interaction effect between map dimensionality and spatial reference frame was not significant, $F(1,60) = 2.141$, $p = 0.149$. Additionally, the three-way interaction effect among navigation mode, map dimensionality, and spatial reference frame was not significant, $F(1,60) = 0.232$, $p = 0.632$.

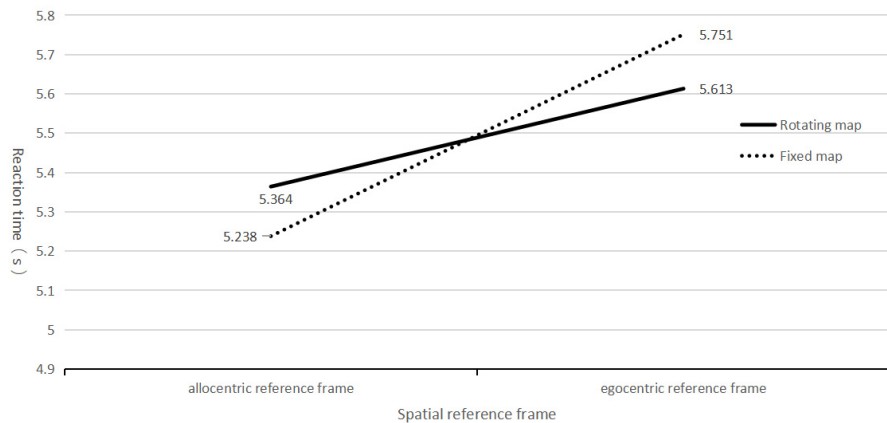

**Figure 7.** Interaction plots of spatial reference frame and navigation mode on reaction time.

*4.2. Cognitive Indicators*

4.2.1. Fixation Duration

Descriptive statistics for the fixation duration are presented in Table 3.

**Table 3.** Descriptive statistics of fixation duration (S).

| Map Dimensionality | Navigation Mode | Spatial Reference Frame | Average Value | Standard Deviation |
|---|---|---|---|---|
| 2D map | Fixed map | Allocentric reference frame | 1.446 | 0.901 |
| | | Egocentric reference frame | 2.023 | 1.130 |
| | | overall mean | 1.730 | 1.052 |
| 2D map | Rotating map | Allocentric reference frame | 1.510 | 0.883 |
| | | Egocentric reference frame | 2.054 | 1.086 |
| | | overall mean | 1.778 | 1.017 |
| 3D map | Fixed map | Allocentric reference frame | 1.217 | 0.982 |
| | | Egocentric reference frame | 2.200 | 1.153 |
| | | overall mean | 1.700 | 1.171 |
| 3D map | Rotating map | Allocentric reference frame | 1.270 | 0.957 |
| | | Egocentric reference frame | 2.005 | 1.030 |
| | | overall mean | 1.631 | 1.053 |

The data were found to meet the criteria of normal distribution and homogeneity of variance ($p > 0.05$). A repeated measures ANOVA conducted revealed the following:

The main effect for spatial reference frame was significant, $F(1,60) = 8.008$, $p = 0.006$, $p < 0.01$. Participants with an allocentric reference frame displayed a significantly shorter fixation duration than those with an egocentric reference frame. The main effect for map dimensionality was not significant, $F(1,60) = 2.386$, $p = 0.128$. Additionally, the main effect for navigation mode was not significant, $F(1,60) = 0.115$, $p = 0.736$.

The interaction effect between map dimensionality and navigation mode was significant, $F(1,60) = 4.126$, $p = 0.047$, $p < 0.05$. Upon conducting a simple effects analysis (Figure 8), it was observed that for the rotating map, there was a significant difference in the fixation duration between the 2D map and 3D map, $p = 0.009$, $p < 0.01$. Specifically, the fixation duration for the 2D map was significantly longer than that for the 3D map. However, for the fixed map, there was no significant difference in the fixation duration between the 2D and 3D maps, $p = 0.711$.

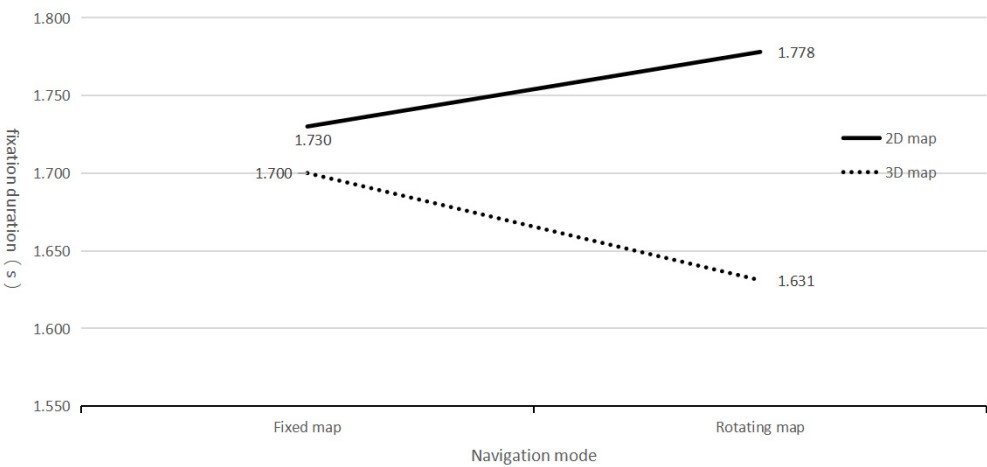

**Figure 8.** Interaction plots of navigation mode and map dimensionality on fixation duration.

The interaction effect between map dimensionality and spatial reference frame was significant, $F(1,60) = 7.239$, $p = 0.009$, $p < 0.01$. A simple effects analysis (Figure 9) revealed that for participants within the allocentric reference frame, there was a significant difference in the fixation duration between the 2D and 3D maps, $p = 0.004$, $p < 0.01$, with the 2D map having a notably longer fixation duration than the 3D map. Conversely, for participants within the egocentric reference frame, there was no significant difference in fixation duration between the 2D and 3D maps, $p = 0.425$.

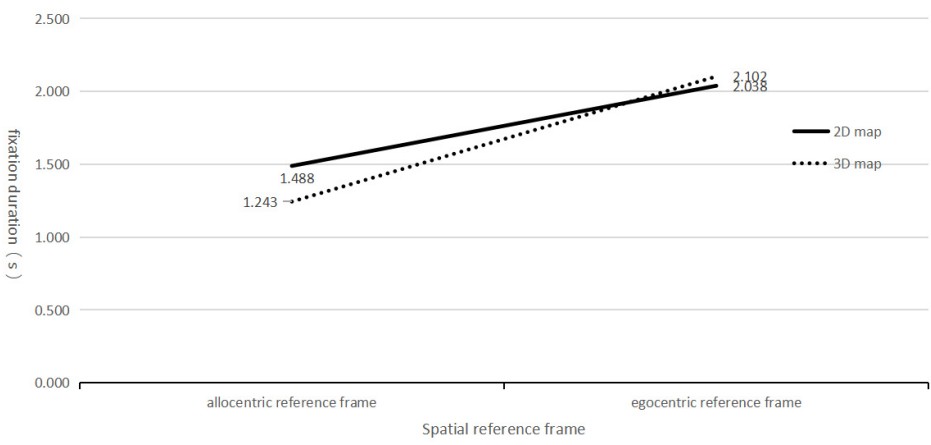

**Figure 9.** Interaction plots of spatial reference frame and map dimensionality on fixation duration.

The interaction effect between navigation mode and spatial reference frame was also significant, $F(1,60) = 4.089$, $p = 0.048$, $p < 0.05$. Further simple effects analysis (Figure 10) indicated that there was a significant difference in fixation duration between participants in the allocentric and egocentric reference frames for the fixed map, $p = 0.004$, $p < 0.01$. participants in the allocentric reference frame exhibited a significantly shorter fixation duration than those in the egocentric reference frame. For the rotating map, a significant difference was also noted between the two reference frames, $p = 0.012$, $p < 0.05$, with the allocentric reference frame again showing shorter fixation durations than the egocentric reference frame.

The three-way interaction among navigation mode, map dimensionality, and spatial reference frame was not significant, $F(1,60) = 3.409$, $p = 0.070$.

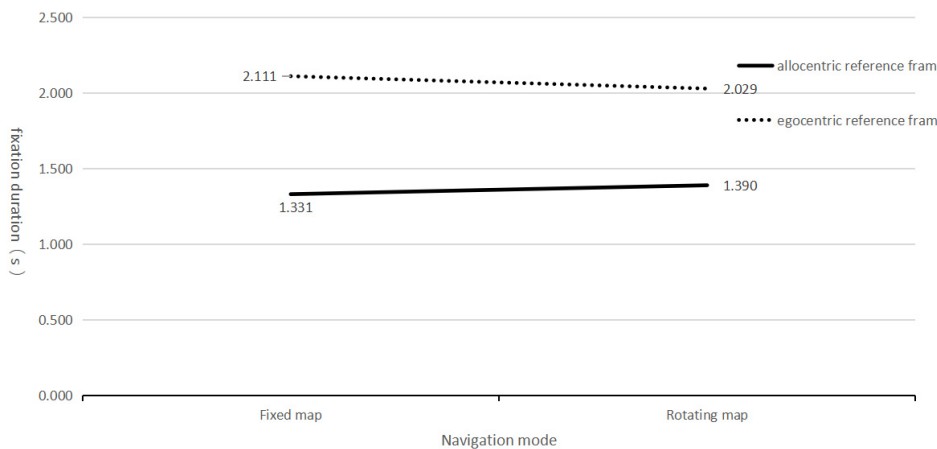

**Figure 10.** Interaction plots of spatial reference frame and navigation mode on fixation duration.

### 4.2.2. Pupil Diameter

Descriptive statistics for the fixation duration are presented in Table 4.

**Table 4.** Descriptive statistics of pupil diameter (mm).

| Map Dimensionality | Navigation Mode | Spatial Reference Frame | Average Value | Standard Deviation |
|---|---|---|---|---|
| 2D map | Fixed map | Allocentric reference frame | 3.088 | 0.306 |
| | | Egocentric reference frame | 3.363 | 0.404 |
| | | overall mean | 3.223 | 0.381 |
| 2D map | Rotating map | Allocentric reference frame | 3.081 | 0.274 |
| | | Egocentric reference frame | 3.396 | 0.378 |
| | | overall mean | 3.236 | 0.363 |
| 3D map | Fixed map | Allocentric reference frame | 3.443 | 0.349 |
| | | Egocentric reference frame | 3.718 | 0.394 |
| | | overall mean | 3.578 | 0.394 |
| 3D map | Rotating map | Allocentric reference frame | 3.443 | 0.366 |
| | | Egocentric reference frame | 3.6593 | 0.38971 |
| | | overall mean | 3.5494 | 0.39033 |

The data were found to meet the criteria of normal distribution and homogeneity of variance ($p > 0.05$). A repeated measures ANOVA conducted revealed the following:

The main effect of map dimensionality was significant, $F(1,60) = 503.250$, $p < 0.001$. Specifically, the pupil diameter for the 2D map was significantly smaller than that for the 3D map. The main effect of the spatial reference frame was also significant, $F(1,60) = 9.241$, $p = 0.004$, $p < 0.01$. Participants within the allocentric reference frame exhibited a significantly smaller pupil diameter compared to those in the egocentric reference frame. However, the main effect of the navigation mode was not significant, $F(1,60) = 0.535$, $p = 0.468$.

The interaction effect between map dimensionality and navigation mode was not significant, $F(1,60) = 1.867$, $p = 0.177$. Similarly, the interaction effect between map dimensionality and the spatial reference frame was not significant, $F(1,60) = 2.814$, $p = 0.099$. The interaction effect between navigation mode and the spatial reference frame was also non-significant, $F(1,60) = 0.169$, $p = 0.682$. Additionally, the three-way interaction among navigation mode, map dimensionality, and the spatial reference frame was not significant, $F(1,60) = 2.496$, $p = 0.119$.

### 4.2.3. Average Saccade Amplitude

Saccade amplitude reflects the breadth of information processed at once by a participant during task completion, thereby indicating the ease of processing the experimental materials. Analysis of the average saccade is also conducted to validate conjectures made based on fixation duration results [33]. Saccade amplitude measures the angular distance between consecutive fixation points. It indicates the breadth of information processed during a task, reflecting the task's complexity. The average saccade amplitude is the total amplitude divided by the number of saccades. Descriptive statistics for the average saccade amplitude are presented in Table 5.

**Table 5.** Descriptive statistics of average saccade amplitude (mm).

| Map Dimensionality | Navigation Mode | Spatial Reference Frame | Average Value | Standard Deviation |
|---|---|---|---|---|
| 2D map | Fixed map | Allocentric reference frame | 3.976 | 1.158 |
|  |  | Egocentric reference frame | 4.067 | 1.170 |
|  |  | overall mean | 4.021 | 1.155 |
| 2D map | Rotating map | Allocentric reference frame | 3.876 | 1.101 |
|  |  | Egocentric reference frame | 3.993 | 1.513 |
|  |  | overall mean | 3.934 | 1.310 |
| 3D map | Fixed map | Allocentric reference frame | 3.062 | 0.955 |
|  |  | Egocentric reference frame | 3.196 | 0.903 |
|  |  | overall mean | 3.128 | 0.925 |
| 3D map | Rotating map | Allocentric reference frame | 3.384 | 1.418 |
|  |  | Egocentric reference frame | 3.383 | 1.008 |
|  |  | overall mean | 3.384 | 1.223 |

The data were found to meet the criteria of normal distribution and homogeneity of variance ($p > 0.05$). A repeated measures ANOVA conducted revealed the following:

The main effect of map dimensionality was significant, $F(1,60) = 55.821$, $p < 0.001$. The average saccade amplitude for 2D maps was significantly larger than for 3D maps. The main effect of navigation mode was not significant, $F(1,60) = 0.781$, $p = 0.380$. The main effect of spatial reference frame was also not significant, $F(1,60) = 0.117$, $p = 0.733$.

The interaction effect between map dimensionality and navigation mode was not significant, $F(1,60) = 3.117$, $p = 0.083$. Similarly, the interaction effect between map dimensionality and spatial reference frame was not significant, $F(1,60) = 0.038$, $p = 0.846$. The interaction effect between navigation mode and spatial reference frame was not significant, $F(1,60) = 0.081$, $p = 0.777$. Lastly, the three-way interaction effect among navigation mode, map dimensionality, and spatial reference frame was not significant, $F(1,60) = 0.172$, $p = 0.680$.

### 5. Discussion

Based on the data and differences observed in map dimensionality and navigation modes within different spatial reference frames, our study provides comprehensive analysis results. The key findings are as follows:

1.  Accuracy rate: In the map navigation task, using a 2D fixed map in the allocentric reference frame resulted in a high accuracy rate with a low cognitive load and a low reaction time. Similarly, using a 2D rotating map also led to a high accuracy rate, a low cognitive load, and a low reaction time compared to the reference frame.
2.  Cognitive load and positioning efficiency: The simplicity of 2D maps proved to reduce the cognitive load of users and improve their positioning efficiency more effectively than the complexity of 3D maps in the map navigation task.

3. Cognitive habits and efficiency: The fixed map was found to align better with the cognitive habits of participants in the allocentric reference frame, while the rotating map was more suited to the cognitive habits of participants in the egocentric reference frame. This alignment resulted in improved cognitive efficiency for both groups.

These findings provide valuable insights into the relationships among map design elements, spatial reference frames, and cognitive efficiency in map navigation tasks.

### 5.1. Behavior Indicators

#### 5.1.1. Accuracy Rate

In the experiment, the main effect of navigation mode was significant, with the accuracy rate of the fixed map being significantly lower than that of the rotating map. This finding aligns with previous research, indicating that after completing psychological rotation, spatial judgments may deviate from actual spatial orientation [22]. This suggests that the mental adjustment required for the fixed map introduces additional cognitive load and affects the accuracy of spatial judgments.

The interaction between map dimensionality and spatial reference frame was found to be significant. For participants with an allocentric reference frame, the accuracy rate of using a 3D map was significantly lower than that of using a 2D map. This can be attributed to the increased complexity and altered perspective of the 3D map, which requires additional mental processing and transformation. As a result, the mental representation of the 3D map formed through cognitive mapping may introduce distortion and deformation, leading to a decrease in the accuracy rate. On the other hand, for participants with an egocentric reference frame, there was no significant difference in the accuracy rate between 2D and 3D maps. This indicated that egocentric reference frame participants relied more on self-reference rather than map environment cues, making them less sensitive to changes in map dimensionality [36].

Additionally, the interaction between navigation mode and spatial reference frame was also significant. For participants with an allocentric reference frame, there was no significant difference in the accuracy rate between fixed and rotating maps, with a slightly lower accuracy rate for the fixed map. This suggests that allocentric reference frame participants, who maintain a fixed reference frame based on cardinal directions, are less affected by map rotation. In contrast, for participants with an egocentric reference frame, the accuracy rate of the fixed map was significantly lower than that of the rotating map. This can be attributed to the fact that egocentric reference frame participants relied on self-reference for spatial judgments, and the consistent self-reference frame in the rotating map facilitated their positioning.

#### 5.1.2. Reaction Time

In the experiment, the main effects of map dimensionality and spatial reference frame were found to be significant. The response time for 2D maps was significantly lower than that for 3D maps, and the response time for participants with an allocentric reference frame was significantly lower than that for participants with the egocentric reference frame. Previous research suggested that participants with an allocentric reference frame exhibit stronger mental rotation ability compared to those with the egocentric reference frame, resulting in lower response times for the allocentric reference frame [30].

The interaction between navigation mode and spatial reference frame was found to be remarkable. For the fixed map, the response time of participants with an egocentric reference frame was significantly higher than that of participants with an allocentric reference frame. Similarly, for the rotating map, the response time of participants with an egocentric reference frame was significantly higher than that of participants with an allocentric reference frame. This indicated that the navigation efficiency was higher for participants with an allocentric reference frame using the fixed map, while participants with the egocentric reference frame showed higher efficiency using the rotating map.

### 5.1.3. Behavior Indicators Overview

The concept of mental rotation and target search processes in spatial orientation tasks has been studied by researchers like Shu Mou et al. [18]. When the reference frame of the map matches the self-reference frame, users can bypass the mental rotation process, thereby enhancing the efficiency of self-positioning tasks, known as the adjustment effect [37]. The fixed map, which maintains a consistent reference frame aligned with the cognitive habits of participants with an allocentric reference frame, enables quick judgments when the moving point rotates. On the other hand, the inconsistent spatial reference frame before and after the rotation of the rotating map leads to cognitive differences for participants with an allocentric reference frame, requiring more time. In contrast, participants with an egocentric reference frame, who adopted a self-centered perspective, experienced longer mental rotation when the moving point of the fixed map rotated. The rotating map effectively reduced their mental rotation time. Therefore, the navigation efficiency was higher for participants with an allocentric reference frame using the fixed map, while participants with the egocentric reference frame showed higher efficiency using the rotating map.

The interaction between map dimensionality and navigation mode was also remarkable. For fixed maps, the response time was significantly lower for 2D maps compared to 3D maps. In the case of rotating maps, there was no significant difference in response time between 2D and 3D maps, with slightly lower response time for 2D maps. This indicated that 2D fixed maps and 2D rotating maps exhibited the lowest response time and highest navigation efficiency. The simplicity of 2D maps reduced the need for mental rotation and target search, thereby enhancing navigation efficiency. While fixed maps require additional mental rotation compared to rotating maps, the response time for 2D rotating maps did not significantly differ from that of 2D fixed maps [34]. When combined with the interaction results of navigation mode and spatial reference frame, it is speculated that the improvements of fixed maps for allocentric reference frame participants and rotating maps for egocentric reference frame participants canceled out each other, resulting in insignificant differences in response time between rotating maps and fixed maps.

### 5.2. Cognitive Processing

#### 5.2.1. Fixation Duration

The fixation duration refers to the sum of the fixation time of the participants in all areas of interest on the map, which can objectively reflect the allocation and consumption of the participants' cognitive resources and also reflect the users' cognitive preferences [38]. In the experiment, the main effect of spatial reference frame was significant, and the fixation duration of the allocentric reference frame was significantly shorter than that of the egocentric reference frame. It is speculated that compared with the allocentric reference frame, the participants relied on their own position with the environment and buildings to identify when turning, so they needed more cognitive resources and a longer fixation duration.

The interaction between map dimensionality and navigation mode is remarkable. For the fixed map, there was no significant difference in the fixation duration between the 2D map and the 3D map; for rotating map, the fixation duration of the 2D map was significantly higher than that of the 3D map; for the 2D map, there was no significant difference between the fixed map and the rotating map; for the 3D map, there was no significant difference between the fixed map and the rotating map. It is speculated that compared with the fixed map, the background of the rotating map changed greatly when the moving point turned, which attracted more attention of the participants, and it was not because the rotating map was difficult to recognize that it led to the increase of fixation duration.

The interaction between map dimensionality and spatial reference frame was significant, and the fixation duration of the participants using the 2D map navigation in the allocentric reference frame was significantly longer than that of participants using 3D map navigation, and there was no significant difference between the fixation duration of the participants using the two kinds of map navigation in the egocentric reference frame. Research

conjectures are limited by the change in the field of view. In the process of transforming a 2D map into a 3D map, the area presented by the same geographical information in the map becomes smaller, and the cognitive processing area of the participants becomes smaller, which leads to a decrease in fixation duration. However, participants with an egocentric reference frame exhibited longer fixation durations on maps of any dimensionality, possibly due to factors such as mental rotation, compared to those with an allocentric reference frame. Even if the area of the 3D map was smaller than that of the 2D map, there was still no significant difference in the fixation duration between the participants in the egocentric reference frame.

The interaction between navigation mode and spatial reference frame was significant. For participants with an allocentric reference frame, the fixation duration on fixed maps was slightly shorter than on rotating maps. In contrast, for those with an egocentric reference frame, the fixation duration on static maps was marginally longer than that on rotating maps, though the difference was not statistically significant. However, the cognitive load for allocentric participants using different navigational maps was significantly lower than for those with an egocentric reference frame. This suggested that while matching an individual's spatial reference frame with the corresponding navigation mode (e.g., egocentric participants using rotating maps) can reduce cognitive load, participants with an egocentric frame still exhibited longer fixation durations, potentially due to factors like mental rotation.

### 5.2.2. Pupil Diameter

Participants employing an allocentric reference frame demonstrated enhanced spatial cognitive performance, likely attributed to the reduced cognitive demands of circumventing the mental rotation process. The simplicity of 2D maps further diminished their cognitive strain. Such superior performance might be rooted in the inherent nature of our experimental tasks—in basic navigational challenges, those with allocentric reference frames often outperformed their counterparts. This is consistent with findings of Brunye et al., who highlighted the advantages of allocentric representations in certain navigational contexts [39].

### 5.2.3. Saccade Amplitude

In the experiment, there was a significant main effect for map dimensionality. The average saccade amplitude for 2D maps was significantly greater than that for 3D maps. This suggested that a single fixation point on a 2D map processes considerably more information than on a 3D map. Compared to 3D maps, 2D maps offer more concise and intuitive geographical information within the same area [40].

### 5.2.4. Cognitive Processing Overview

In general, our research explored the cognitive dynamics of map navigation. This study underscores the vital intersection of user cognition and map design, offering guidance for designers aiming to optimize user experience. The comparative analysis between 2D and 3D maps revealed that overly detailed background information, especially in 3D maps, can lead to increased cognitive load, elongated reaction times, and reduced accuracy. Such findings suggested a paradigm shift toward simplifying and decluttering designs. Removing redundant and irrelevant information and emphasizing key geographical details, like landmarks and nodal points, are foundational steps in enhancing user comprehension and interaction.

The variances between users adopting allocentric versus egocentric reference frames were particularly illuminating. Designers might need to cater to these different cognitive styles by offering flexible map designs or multiple viewing options. This adaptability can facilitate a more personalized and effective navigation experience for users. Recognizing users as co-designers, giving them room to tailor their experiences, and providing immediate and effective feedback mechanisms can enhance user satisfaction and efficiency.

## 6. Conclusions

Overall, this study explored the effects of spatial reference frame, navigation mode, and map dimensionality on spatial orientation efficiency, employing comparative experiments and eye-movement analysis. The results indicate that these three factors significantly impact spatial orientation efficiency and exhibit interactive influences.

The integration of the spatial reference frame provided insights into individuals' map perception and navigation strategies. The study demonstrated that the spatial reference frame, navigation mode, and map dimensionality jointly affect spatial orientation efficiency. The interplay of these factors underscores the intricacy of map interpretation and the cognitive processes involved.

This research's endeavor to replicate the complexity of real-world navigation by incorporating spatial reference frames, map dimensionality, and navigation modes may not fully capture the subtleties of actual map use, due to the controlled lab setting. While offering a comprehensive perspective, this broad approach might obscure the specific impact of each variable. Furthermore, the study did not assess participants' inherent spatial abilities, which could play a crucial role in understanding map-related tasks, as measured by established metrics like the Perspective Taking/Spatial Orientation Test and the Santa Barbara Sense of Direction Scale [28,41].

Acknowledging these limitations, we propose that future research could gain from an in-depth examination of these variables separately to clarify their individual contributions to navigational efficiency. Moreover, incorporating assessments of spatial abilities could provide richer insights into how users interact with maps. This study lays the foundation for such detailed investigations, emphasizing the importance of balancing comprehensive explorations of navigational phenomena with in-depth analysis of its elements.

In summary, this study sheds light on the roles of spatial reference frame, navigation mode, and map dimensionality in spatial orientation efficiency. The findings enhance our understanding of map cognition and offer guidance for the development of more efficient map-navigation tools and services. Continued research is essential to deepen our knowledge of map-usage behaviors and cognitive responses in real-life scenarios.

**Supplementary Materials:** The following supporting information can be downloaded at: https://www.mdpi.com/article/10.3390/ijgi12120476/s1.

**Author Contributions:** Conceptualization, Hongyun Guo; methodology, Hao Fang; software, Zhong Wang; validation, Hao Fang; formal analysis, Zhong Wang; investigation, Hongyun Guo; resources, Nai Yang; data curation, Zhong Wang; writing—original draft preparation, Zhong Wang; writing—review and editing, Hongyun Guo; visualization, Nai Yang; supervision, Nai Yang; project administration, Hongyun Guo; funding acquisition, Nai Yang. All authors have read and agreed to the published version of the manuscript.

**Funding:** This research was funded by the National Natural Science Foundation of China, grant number 42171438, and the Scientific Research Foundation of Wuhan Institute of Technology (K2023064).

**Data Availability Statement:** The data presented in this study are available on request from the corresponding author. The data are not publicly available as they are currently being used for ongoing research.

**Conflicts of Interest:** The authors declare no conflict of interest.

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
