# Peer review of "Effects of Spatial Reference Frames, Map Dimensionality, and Navigation Modes on Spatial Orientation Efficiency"

_ijgi, doi:10.3390/ijgi12120476_

Round 1

Reviewer 1 Report (Previous Reviewer 2)

Comments and Suggestions for Authors

I am particularly happy to see that the authors have taken into account all reviewers' comments and have submitted a much better version of their original manuscript. As I have mentioned in my previous review, this kind of studies are very interesting and when they are carefully designed and executed can provide fruitful insights into how humans use the map and how they deploy their cognitive and spatial skills. In this respect, the study in this current version is much more carefully explained and detailed and the fact that the authors accompany the manuscript with their raw data adds to the significance of their research and results. 

Moreover, the related work section is far more informative in this version than before and seminal work in the field has been included. 

I have spotted some issues that should be tackled with though;

1.  lines 222-225; for the allocentric frame of reference you use cardinal points; with "their heads facing south", while for the egocentric you say: "their heads facing right". The use of different ways to describe both reference systems is somewhat misleading, and to my understanding it describes the two systems wrong. I think the egocentric system is the one far right in figure two which you describe correctly, while the allocentric is the one in the middle which you wrongly describe with the phrase facing south which is north. Again better use upwards/downwards instead of north/south, or left/right.

2.line 240-241: while the navigation mode and Map dimensionality are internal variables of the participants. Is this true?? You may mean independent variables?

3. The term correct rate throughout the paper. I would suggest using correctness rate or even better accuracy rate

4. Figures 4-9 Interaction Effects may better change to Interaction Plots. The same for tables 10-12 but also these are not tables but figures.

5 In section 5.2, lines 586, 597-590, 598; since there is no statistically significance difference, you should not write anything more. 

6. Section 5.2.3 lines 632-637. Maybe this information should move earlier in the text, for instance to section 4.2.3.; it does not belong to the results, it rather describes the variable; saccade amplitude. To be more "tidy" it would be better to describe all variables of the analysis (e.g., accuracy, response time, fixation etc. in section 3 and not in the analysis/results or discussion sections.

7. lines 643-658; I believe summarize section 5.2 therefore it would better if you added another section after 5.2.3, 5.2.4 with a heading/title such as: "cognitive processing overview". The same holds for lines 541-569, which summarize section 5.1 behavior indicators. 

8. lines 678-679 references on these tests/questionnaires are missing

9. avoid capitalizing Map dimensionality since navigation mode or spatial reference frame are not capitalized.

10 Finally, line 380 a reference is missing

Comments on the Quality of English Language

Minor issues with English spotted which I think can be addressed after thoroughly reviewing the paper.

A couple of points may require further attention: 

1. lines 16-17: Conversely compared to the reference frame, participants achieved a higher correct rate, lower cognitive load, and shorter reaction time when using a 2D rotating map. The meaning of the sentence is not very clear

2. lines 87-90: consider rephrasing. Some words may be missing(?) 

3. lines 273-274; Before commencing the experiment, the examiner will provide the participants with relevant instructions and precautions, please consider rephrasing; before the study begins, the examiner provides...

Author Response

First and foremost, we sincerely thank you for taking the time to re-evaluate our manuscript. We deeply appreciate your insightful comments and your recognition of the improvements we made based on the initial review feedback. We are heartened by your acknowledgment that we have diligently considered all reviewers' suggestions and have submitted an enhanced version of our original manuscript. Your encouraging words and validation of the detailed explanations, comprehensive information, and the inclusion of raw data in the current version mean a great deal to us. However, we understand that there remain areas for enhancement. In response to the specific issues you raised, here are our itemized revisions:

Reviewer point #1: lines 222-225; for the allocentric frame of reference you use cardinal points; with "their heads facing south", while for the egocentric you say: "their heads facing right". The use of different ways to describe both reference systems is somewhat misleading, and to my understanding it describes the two systems wrong. I think the egocentric system is the one far right in figure two which you describe correctly, while the allocentric is the one in the middle which you wrongly describe with the phrase facing south which is north. Again better use upwards/downwards instead of north/south, or left/right.

Author response #1: Thank you for your valuable comments and insights regarding the description of the reference systems in our manuscript. We sincerely apologize for any confusion caused by our initial wording. Upon reviewing lines 222-225, as you rightly pointed out, we realized that the use of different descriptors for the allocentric and egocentric frames of reference may lead to ambiguity. Your feedback has been instrumental in helping us identify this inconsistency. In response, we have made the necessary corrections to provide a more consistent and accurate description. Specifically, we have amended the text to use "upwards/downwards" in place of "north/south" to avoid potential confusion and to ensure that our descriptions align with the illustrations provided in figure two.

Modified content:” If a participant consistently recreated the scene by placing the dragonflies with their heads facing upwards, it indicated the use of an allocentric reference frame. If a partic-ipant placed the dragonflies with their heads facing downwards (toward the right) when recreating the scene, it signaled the use of an egocentric reference frame (as il-lustrated in Figure 2).”

Reviewer point #2: line 240-241: while the navigation mode and Map dimensionality are internal variables of the participants. Is this true?? You may mean independent variables?

Author response #2: You've correctly identified a misrepresentation in that section. Indeed, my choice of words while describing the experimental design was not apt. Our intention was to convey that both "navigation mode" and "Map dimensionality" are repeated measures variable, not internal attributes of the participants. I apologize for the oversight and appreciate your correction. I have now revised that portion to accurately reflect that these variables serve as repeated measures variable in our mixed experimental design. Thank you once again for your thorough review and valuable feedback. We highly value and appreciate your guidance, which helps enhance the quality and accuracy of our manuscript.

Modified content:” The spatial reference frame is considered an independent groups variable, while the navigation mode and map dimensionality are repeated measures variable of the participants.”

Reviewer point #3: The term correct rate throughout the paper. I would suggest using correctness rate or even better accuracy rate

Author response #3: Thank you for your valuable feedback on our manuscript. Upon reviewing your comment regarding the term "correct rate", I understand and agree with your perspective. "Accuracy rate" is indeed a more appropriate and standard term within the research community. I have revised the manuscript to replace "correct rate" with "accuracy rate" throughout, ensuring consistency and clarity. Your suggestion greatly contributes to enhancing the quality and precision of our work.

Reviewer point #4: Figures 4-9 Interaction Effects may better change to Interaction Plots. The same for tables 10-12 but also these are not tables but figures.

Author response #4: I sincerely apologize for the oversight in our manuscript, and thank you for your meticulous review and invaluable feedback. Based on your suggestion, we have revised our manuscript and changed the titles of Figures 4-9 from "Interaction Effects" to "Interaction Plots". Additionally, the oversight regarding Tables 9-10 has been rectified and they are now correctly labeled as figures.

Reviewer point #5: In section 5.2, lines 586, 597-590, 598; since there is no statistically significance difference, you should not write anything more.

Author response #5: Thank you for your thoughtful feedback regarding our manuscript. Upon revisiting section 5.2, I concur with your observation. We have now streamlined the content in lines 586, 597-590, and 598 to ensure clarity, removing any redundant information associated with the non-significant results. We strive to maintain the rigor and relevance of our findings, and your guidance has been instrumental in this regard.

Modified content:” for 3D map, there is no significant difference between fixed map and rotating map; It is speculated that compared with the fixed map, the background of the rotating map changed greatly when the moving point turned, which attracted more attention of the participants, and it was not because the rotating map was difficult to recognize that it led to the increase of Fixation duration.

The interaction between map dimensionality and spatial reference frame is significant, and the fixation duration of the participants using 2D map navigation in allocentric reference frame is significantly longer than that of 3D map navigation, and there is no significant difference between the fixation duration of the participants using two kinds of map navigation in egocentric reference frame.”

Reviewer point #6: Section 5.2.3 lines 632-637. Maybe this information should move earlier in the text, for instance to section 4.2.3.; it does not belong to the results, it rather describes the variable; saccade amplitude. To be more "tidy" it would be better to describe all variables of the analysis (e.g., accuracy, response time, fixation etc. in section 3 and not in the analysis/results or discussion sections.

Author response #6: Thank you for your insightful suggestion to improve our manuscript. Based on your recommendation, we have moved the information from Section 5.2.3 (lines 632-637) to Section 4.2.3. You are absolutely right; this information primarily describes the variable "saccade amplitude" rather than actual results. To ensure a more organized structure for our paper, we have now detailed all variables for the analysis (e.g., accuracy, response time, fixation, etc.) within Section 3.3, refraining from discussing them within the analysis/results or discussion sections.

Modified content:” Specifically, the accuracy rate represents the probability of participants correctly completing the navigational positioning task, determined by collecting mouse clicks within a predefined area corresponding to an arrow. The reaction time refers to the time taken by participants to complete the navigation and positioning task. Fixation duration refers to the total time that a participant gazes at all Areas of Interest on a map. It provides an objective measure of the cognitive resources expended by the participant. The pupil diameter, as a measure of cognitive load, can indicate the level of cognitive effort exerted by the participants. Saccade amplitude refers to the angle of eye movement from one fixation point to another. Average saccade amplitude is calculated as the total saccade amplitude divided by the number of saccades [8,33-35].”

Reviewer point #7: lines 643-658; I believe summarize section 5.2 therefore it would better if you added another section after 5.2.3, 5.2.4 with a heading/title such as: "cognitive processing overview". The same holds for lines 541-569, which summarize section 5.1 behavior indicators.

Author response #7: Thank you for your in-depth feedback on the structure of our paper. Based on your suggestions, we have made structural adjustments to the content. Specifically, we have added the titles "Behavior Indicators Overview" for section 5.1.3 and "Cognitive Processing Overview" for section 5.2.4. This reorganization helps summarize and clarify the main findings of sections 5.1 and 5.2, ensuring a more logical presentation of our content. These changes play a crucial role in enhancing the clarity and structure of the paper, offering our readers a more coherent narrative.

Reviewer point #8: This makes no sense to me. What are external visual cues in this context? The participants had only the map, correct? I don’t see what external cues they could be relying upon.

Author response #8: Thank you for pointing out the missing references in lines 678-679. We have now addressed this by including the pertinent references for the two tests, the Perspective Taking Test and the Santa Barbara Sense of Direction Scale. Specifically, we added the following references:

41.Hegarty, M.; Waller, D. A dissociation between mental rotation and perspective-taking spatial abilities. Intelligence 2004, 32, 175-191.

42.Hegarty, M.; Richardson, A.E.; Montello, D.R.; Lovelace, K.; Subbiah, I. Development of a self-report measure of environmental spatial ability. Intelligence 2002, 30, 425-447.

Reviewer point #9: avoid capitalizing Map dimensionality since navigation mode or spatial reference frame are not capitalized.

Author response #9: Thank you for pointing out this inconsistency in our manuscript. We have revised the text and ensured that the capitalization is consistent throughout the document. Now, "Map dimensionality" is written in lowercase, aligning with the formatting of "navigation mode" and "spatial reference frame". We appreciate your attention to detail, as it helps in maintaining the professionalism and readability of our paper.

Reviewer point #10: Finally, line 380 a reference is missing.

Author response #10: Thank you for pointing out the oversight in our manuscript. You are right, and it was an oversight on our part. We have now added the appropriate reference to line 380:

34.Duchowski, T.A. Eye tracking: methodology theory and practice; Springer, 2017, ISBN 3319578839.

In response to the reviewer's comments regarding the quality of English in our manuscript, we have made the following revisions:

Reviewer point #1: Finally, line 380 a reference is missing.

Author response #1: Thank you for pointing out the ambiguity in lines 16-17. We realize that the phrasing might be confusing. We have now reworded the sentence to better convey the information and avoid any misunderstandings.

Modified content:” When using a 2D fixed map in an allocentric reference frame, participants exhibit a high correct rate, low cognitive load, and short reaction time. In contrast, when operating within an egocentric reference frame using a 2D rotating map, participants demonstrate a higher correct rate, reduced cognitive load, and quicker reaction time.”

Reviewer point #2: lines 87-90: consider rephrasing. Some words may be missing(?)

Author response #2: Thank you for pointing out the issue in lines 87-90. Upon revisiting this section, we indeed identified areas where the phrasing was not smooth. We have now reworded this portion to ensure clarity and coherence in our expression.

Modified content:” Based on previous studies, we've found that in map navigation tasks, participants using 3D maps can be more helpful in quickly searching for required information in complex scenarios. However, 2D maps impose a lower cognitive load, allowing participants to demonstrate greater spatial cognitive efficiency in simpler tasks.”

Reviewer point #3: lines 273-274; Before commencing the experiment, the examiner will provide the participants with relevant instructions and precautions, please consider rephrasing; before the study begins, the examiner provides...

Author response #3: We concur with your perspective and have revised the sentence as per your guidance. This indeed makes the statement more concise and clear.

Modified content:” Before the study begins, the examiner provides the participants with relevant instructions and precautions.”

Thank you once again for your dedication and invaluable suggestions. We sincerely appreciate the thorough guidance you've provided throughout this process. With your recommendations, we hope that our manuscript has been further refined. We believe these revisions have elevated the quality of the manuscript and hope it will make a meaningful contribution to the field.

Reviewer 2 Report (Previous Reviewer 3)

Comments and Suggestions for Authors

I have no further comments.

Author Response

We would like to express our sincere gratitude to you for the time and effort dedicated to reviewing our manuscript. We are pleased to learn that no further revisions have been suggested after the modifications we implemented following the first round of reviews. We appreciate the thorough reviews and constructive comments that have undoubtedly improved the quality of our manuscript. We confirm that we have carefully considered and addressed the feedback provided in the initial review, and we are hopeful that the manuscript is now ready for publication.

Reviewer 3 Report (New Reviewer)

Comments and Suggestions for Authors

Thank you for submitting the paper.

The scope of the paper is very wide and not focused. Although good efforts have been made to connect different sections. However, the paper lacks in clarity and completeness. It is suggested to measure/explore the impact of one variable instead of three i.e. spatial reference frames, Map dimensionality, and navigation modes on spatial orientation efficiency. The paper may be considered if scope is narrowed down and written with clarity and in its totality. Some observations are as the following:

·        Please change “Navigation maps” to “Navigational maps” throughout the paper

·        Line 37: One would expect the definition of “map dimensionality”. Please add that.

·        It is not clear that the authors are referring to paper or digital maps

·        Line 70-77: References are not properly cited. E.g. Partala et al., Lei et al. It is suggested to cite as Partala et al.[14], Lei et al. [15]

·        Line 88: The statement, “…….. 3D maps can search for information more accurately…..” needs to be rephrased. For example, 3D maps can be more helpful in quickly searching for required information …..

·        Line 92: Rephrase the statement, “Various studies have demonstrated that different navigation methods have distinct effect on…….”.  For example, various studies have demonstrated that different navigation methods such as GPS, compass and maps etc have distinct effect on…..

Comments on the Quality of English Language

Difficult to understand. Needs extensive editing.

Author Response

First and foremost, we sincerely thank you for the invaluable feedback and suggestions on our manuscript. Your insights have been tremendously helpful, prompting us to reflect more deeply on the structure and content of our article.

We acknowledge the reviewer's concerns regarding the broad scope of our study. The ambition of our paper was to reflect the multifaceted nature of navigational practices in real-world settings, where map dimensionality, navigation modes, and spatial reference frames are inherently intertwined. However, we recognize that this comprehensive approach might have introduced complexities that obscured the singular impact of each variable.

The interdependency among the variables, while reflective of genuine user experiences, does pose challenges in isolating individual effects. Consequently, this could affect the clarity and perceived completeness of the study. We agree that focusing on one variable at a time could enhance the specificity of our findings and provide a more straightforward narrative.

In future work, we plan to dissect these complex interactions through a series of more narrowly focused studies, which will allow us to delineate the distinct influence of each variable on spatial orientation efficiency. Meanwhile, we have revised our manuscript to better highlight the interplay between each variable and the positioning efficiency of maps.

We also concede that a single-variable approach would facilitate a clearer statistical analysis and could be more easily digested by our readers. Nevertheless, we believe that our current multi-variable approach brings forward a comprehensive overview that could serve as a valuable reference for subsequent, more granular research efforts.

In light of these reflections, we have expanded our discussion to explicitly address these limitations, and we have underscored the potential for future research to build on our findings with more focused investigations.

Regarding your other suggestions, we have provided the following detailed responses:

Reviewer point #1: Please change “Navigation maps” to “Navigational maps” throughout the paper.

Author response #1: Thank you for pointing out this detail. We deeply appreciate your thorough review and valuable suggestions for improving our manuscript. In line with your recommendation, we have revised "Navigation maps" to "Navigational maps" throughout the paper. Your feedback is invaluable to us. Once again, thank you for your constructive comments.

Reviewer point #2: Line 37: One would expect the definition of “map dimensionality”. Please add that.

Author response #2: We sincerely appreciate your valuable feedback. Your meticulous and professional attention to the definition of "map dimensionality" has undoubtedly enhanced the rigor of our research. Indeed, a clear definition of this term is pivotal for the coherence of the manuscript and the understanding of readers. Following your suggestion, we have added a definition for "map dimensionality" in the relevant section to ensure clarity for our readers.

Modified content:” Map dimensionality refers to the spatial representation on a navigational map. Specifically, it indicates whether the map portrays information in two dimensions (2D) or three dimensions (3D)[3]. Based on this dimensionality, navigational maps can be classified into 2D and 3D types.”

Reviewer point #3: It is not clear that the authors are referring to paper or digital maps.

Author response #3: Thank you very much for pointing out this ambiguity. Your keen observation ensures that our work remains precise and understandable to our readers. We would like to clarify that we are referring to digital maps in our study. To clarify and eliminate any ambiguity, we have provided an explanation in section 1 of the manuscript. We truly value your insightful feedback and believe it significantly enhances the quality of our work.

Modified content:” Building upon existing research, this article introduces spatial reference systems as an individual characteristic factor in digital map navigation. It delves into the interplay between spatial reference frames, map dimensionality, and navigation mode concerning spatial orientation efficiency. Through empirical research, this study offers design principles to inform future navigational map design practices.”

Reviewer point #4: Line 70-77: References are not properly cited. E.g. Partala et al., Lei et al. It is suggested to cite as Partala et al.[14], Lei et al. [15]

Author response #4: We apologize for any oversight in the citation style. To address this issue, we have conducted a thorough review of the entire manuscript and have updated the references accordingly. For instance, we have revised the references as Partala et al. [14] and Lei et al. [15] to ensure proper citation format throughout the paper. Your guidance has contributed to improving the overall quality of our manuscript, and we are grateful for your constructive suggestions. Thank you for being so diligent in reviewing our work.

Reviewer point #5: Line 88: The statement, “…….. 3D maps can search for information more accurately…..” needs to be rephrased. For example, 3D maps can be more helpful in quickly searching for required information …..

Author response #5: We appreciate your suggestion and have rephrased the sentence to improve clarity and accuracy of expression. Thank you for your guidance and attention, which contribute to enhancing the quality of our manuscript.

Modified content:” Based on previous studies, we've found that in map navigation tasks, participants using 3D maps can be more helpful in quickly searching for required information in complex scenarios. However, 2D maps impose a lower cognitive load, allowing participants to demonstrate greater spatial cognitive efficiency in simpler tasks.”

Reviewer point #6: Line 92: Rephrase the statement, “Various studies have demonstrated that different navigation methods have distinct effect on…….”.  For example, various studies have demonstrated that different navigation methods such as GPS, compass and maps etc. have distinct effect on…..

Author response #6: Thank you for your suggestion. We have rephrased the sentence to explicitly list various navigation methods based on the theme of the article. This modification makes the sentence clearer and better reflects our point of view.

Modified content:” Various studies have shown that navigation methods, specifically fixed and rotating maps, have distinct effects on the cognitive efficiency of map navigation.”

Your feedback on the language quality is greatly appreciated. We recognize the need for further editing to enhance the comprehensibility of the paper. Extensive editing has been performed to improve its expression and clarity, ensure that readers can more easily understand the content of the article.

Round 2

Reviewer 3 Report (New Reviewer)

Comments and Suggestions for Authors

Thank you for addressing my concerns.

Comments on the Quality of English Language

Minor editing required

Author Response

Thank you for your valuable feedback and for acknowledging the revisions made in response to your concerns. We greatly appreciate your guidance throughout this process.

Regarding your comments on the quality of the English language used in our manuscript, we have undertaken another thorough review and editing to ensure the clarity and precision of our language. Minor linguistic adjustments have been made to enhance the overall readability and coherence of the text, aligning with academic standards.

We hope these additional revisions meet your expectations and contribute to the overall quality of the manuscript. We are grateful for your attention to detail and constructive suggestions, which have significantly improved our work.

Thank you again for your support and invaluable contributions to our research.

This manuscript is a resubmission of an earlier submission. The following is a list of the peer review reports and author responses from that submission.

Round 1

Reviewer 1 Report

Comments and Suggestions for Authors

Thank you for the opportunity to read this manuscript. It reports the findings of an empirical investigation of the interactions between spatial reference frames and the dimensionality of map features (2D vs 3D) in supporting spatial orientation A factorial experiment has been designed with a mix of within and between subjects factors. The analytical approach (multi-way ANOVA) is generally appropriate. Examining this set of factors should provide some interesting information about how maps can support navigation and way finding tasks. I appreciate that the authors identified the key findings and presented them in a concise fashion in both the abstract and the discussion.

  1. The authors could do a better job of making precise claims. Many of the claims made in the introduction would be correct for maps designed to support navigation, but would not necessarily be correct for other types of maps. Yet all maps are lumped together according to your argument. See below for additional places in the manuscript where imprecise claims or claims with insufficient evidence appear.
  2. I liked that the authors listed several null hypotheses they were trying to test in this work. However, what would be more informative and useful would be to not only hypothesize that two conditions might produce significantly different result, but also to specify the direction of hypothesized difference. Which condition do you expect to perform better than another and why?
  3. Has Levinson’s test been validated for vista scales? It seems to be a test developed for the object scale. How do you know that it also applies to rotation of landscapes in a person’s mental map? Could the adopted reference frame be situationally dependent? In Figure 2 is the smiley-face part of the set of objects the participant is supposed to place? If so, why isn’t it included on the second example?
  4. More information needs to be provided about the eye tracking data collection and analysis. Please see the Dunn et al 2023 paper in the journal Behavior Research methods for the minimum information that should be reported with any eye tracking analysis. You also need to clearly describe how you defined the AOIs for your gaze analysis. You should also cite references that describe what can be inferred from a given eye tracking metric (e.g., fixation duration, pupil diameter).
  5. Who were your participants? What were their ages and gender? Did you test their spatial abilities, for example, using the perspective taking test, or something like the Santa Barbara Sense of Direction Scale? If you did not, that is a limitation of this research. It’s clear that the individual’s spatial abilities will have an influence on their ability to undertake your experimental task and this is a variable that should optimally have been measured and controlled for in your experiment. It seems incorrect to me to refer to adoption of reference frame as a preference - it’s highly likely that the spatial reference frame a participant adopts is related to their underlying spatial abilities rather than a preference. 
  6. There are far too many tables and much of the information communicated in the tables would be more succinctly communicated in line graphs. Also, I don’t understand what “amount to” means in those tables. Is this the overall mean across both spatial reference frames? There is an error in Table 8 where 3D map is listed under the spatial reference frame instead of absolute reference frame. Line graphs will show the main effects and interactions much more clearly than do the tables and make it easier for the reader to digest the many statistical tests you undertook in analyzing your data. The authors might find this webpage helpful in providing examples of how these graphs should look: https://onlinestatbook.com/2/analysis_of_variance/tests_supplementing.html
  7. 3D maps present problems with occlusion, which can make it hard to see the position of the blue dot. This is also probably contributing to why they were harder to use for many participants given that they didn’t have the ability (or at least I believe this is true from your experiment description) to adjust the viewpoint when examining the 3D display. If they could manipulate the viewpoint, this needs to be clearly stated in manuscript.
  8. Line 480: This makes no sense to me. What are external visual cues in this context? The participants had only the map, correct? I don’t see what external cues they could be relying upon.
  9. Lines 561-564: This sentence makes no sense to me. Are you referring to the effect of perspective in 3D views? I don’t see how that would affect gaze duration. Also, I thought you were measuring fixation duration, not gaze duration. These are two different eye tracking metrics. Moreover, your argument here makes no sense to me. You are making claims about spatial cognitive abilities, but you have not measured this about your participants. Or if you have, you haven’t reported this in your manuscript. See point #5 above.
  10. You sometimes make claims without evidence, but actually have the data to provide the relevant evidence. For example, lines 600-602: You could examine whether participants with the relative frame refer to buildings more - the eye tracking scan paths can show you that by counting the number fixations on buildings.
  11. It might be more clear to use the term ‘map feature dimensions’ instead of ‘map dimensions’. I would also recommend the use of the appropriate terminology (egocentric and allocentric) to refer to spatial reference frames. You should be citing Dan Montello’s foundational work on egocentric and allocentric reference frames and route vs survey learning of environments. Alex Klippel’s research on You-Are-Here maps is also relevant to your work and should be cited. I think a more appropriate term for “map positioning” would be “spatial orientation”. Spatial orientation refers to the ability to identify the position or direction of objects or points in space and that is exactly what your experimental task is asking participants to do.
  12. Some of the choices of references are very odd. For example, reference 6 does not investigate positioning efficiency at all. Each reference should be carefully checked for its applicability and relevance to the idea it is being used to support. Furthermore, references could be used more effectively to support the authors’ claims. For example, the authors state (line 92): “However certain studies…” which studies? Finally all author references in the text should be carefully checked. On line 76, you refer to a paper by Robbi Sluter, and you use reference [15], but if I examine there reference list, this paper is not by Robbi Slater… Your reference to [26] in line 532 is also strange. This study did not examine map rotation as you claim it has done. Also, some references such as 27 or 28 it is not clear what these documents are or if they are peer-reviewed scientific literature. Are they student theses? If so, the level should be provided. It would be preferable to cite peer-reviewed literature published in journals accessible to the international community.
  13. Line 106: I don’t know what you mean by ‘inflection point’.  This term needs explanation.
  14. Line 198: “shown simultaneously in the four corners of the map…” — that’s not what figure 2 shows. I see the red dot in only one of the four corners of each map. I also don’t understand what you mean by the blue dot rotating concurrently. Concurrently with what? That whole sentence needs to be rewritten so it’s understandable.
  15. It might also be helpful to draw some implications for map design practice in your discussion of the findings. What should map designers keep doing or do differently as a result of what you found in your study?
Comments on the Quality of English Language

Generally the manuscript is written in idiomatic English. I think most of the language problems I have identified in my review stem from a lack of conceptual clarity rather than English language expression. I have noted some specific things I think are problematic in this regard.

Reviewer 2 Report

Comments and Suggestions for Authors

The authors present the results of a study that explores the impact of spatial reference frames, map dimensions, and navigation modes on map positioning efficiency, as well as their interactions, through empirical eye movement experiments. Overall 60 participants were tested on their correctness and time response in map positioning split into two groups (using either an absolute or a relative spatial reference frame). Eye movement indicators such as gaze duration (fixation) and pupil diameter were registered and compared.

Although this kind of studies are very interesting and when they are carefully designed and executed can provide fruitful insights into how humans use the map and how they deploy their cognitive and spatial skills, this particular study could also be very useful to scholar with similar interests, to map makers and the general public, their is a very important piece missing and this is the raw data that the study provided to the authors. Meaning that I strongly advocate that the authors should provide their collected data so that any interested party can reproduce the study and verify the results. That is why I have rated the quality of presentation and the scientific soundness of the paper as low.

Another major issue, is the missing bibliography in terms of many studies in the field of spatial skills and abilities related to the authors' research question such as mental rotation, perspective taking etc. Thus, section 2 should include many more references of research and studies on perspective taking and mental rotation by Hegarty, Newcombe, Golledge, Montello, Uttal and many others with very significant results on participants' skills especially since many research efforts and empirical studies in the field have high number of participants that validate their results further. Additionally, the references 24 and 25 that are used for the animal arrangement paradigm are wrong; first author is Levinson not Levine.

I also have an issue with the term map dimension. I would suggest that the authors use map dimensionality over map dimension(s), since the latter may refer to the size of the map...In this respect, in line 152: "H2: In the map navigation task, there are significant differences in map navigation efficiency among different map dimensions", should be rephrased into "H2: In the map navigation task, there are significant differences in map navigation efficiency between 2D and 3D maps"

Moreover, I detected a couple of other minor issues such as;

in Table 2, second column; degree of freedom instead of freedom (the same for Tables 9, 17, and 26).

regarding reaction time; please mention the units, i.e., minutes, seconds, milliseconds etc.

Overall the paper, is well structured and organized and the results are well explained and clearly respected, nonetheless, the lack of reproducibility potential and the subsequent verification of results prevents me from suggesting publication.

Comments on the Quality of English Language

No major issues in the use of English language are spotted. Only the miss-understanding regarding the term map dimension that may refer to the map's size. I have a comment in general comments section where I suggest an improvement.

Reviewer 3 Report

Comments and Suggestions for Authors

In general, the manuscript is well written and the methods and results are presented clearly and in a straightforward way. However, too many tables make the paper lack readability and continuity. I recommended the author consider using more graphical forms to express the results.